# SD-HRNet: Slimming and Distilling High-Resolution Network for Efficient Face Alignment

**DOI:** 10.3390/s23031532

**Published:** 2023-01-30

**Authors:** Xuxin Lin, Haowen Zheng, Penghui Zhao, Yanyan Liang

**Affiliations:** 1Zhuhai Da Heng Qin Technology Development Co., Ltd., Zhuhai 519000, China; 2Faculty of Innovation Engineering, Macau University of Science and Technology, Macau 999078, China

**Keywords:** face alignment, knowledge distillation, network pruning, lightweight model

## Abstract

Face alignment is widely used in high-level face analysis applications, such as human activity recognition and human–computer interaction. However, most existing models involve a large number of parameters and are computationally inefficient in practical applications. In this paper, we aim to build a lightweight facial landmark detector by proposing a network-level architecture-slimming method. Concretely, we introduce a selective feature fusion mechanism to quantify and prune redundant transformation and aggregation operations in a high-resolution supernetwork. Moreover, we develop a triple knowledge distillation scheme to further refine a slimmed network, where two peer student networks could learn the implicit landmark distributions from each other while absorbing the knowledge from a teacher network. Extensive experiments on challenging benchmarks, including 300W, COFW, and WFLW, demonstrate that our approach achieves competitive performance with a better trade-off between the number of parameters (0.98 M–1.32 M) and the number of floating-point operations (0.59 G–0.6 G) when compared to recent state-of-the-art methods.

## 1. Introduction

Face alignment, also known as facial landmark detection, aims at locating a set of semantic points on a given face image. It usually serves as a critical step in many face applications, such as face recognition [1], expression analysis [2], and driver-status tracking [3], which are significant components of human–computer interaction systems. As an example, face alignment is used to generate a canonical face in the preprocessing of face recognition [4,5]. In the past decade, there have been many methods and common datasets reported in the literature [6,7,8,9,10,11,12,13,14,15,16,17,18,19,20,21,22] to promote the development of face alignment. Nevertheless, it remains a challenging task to develop an efficient and robust facial landmark detector that performs well in various unconstrained scenarios.

In early works, the methods [6,7,8,9] based on cascaded regression made significant progress on face alignment. They could learn a mapping function to iteratively refine the estimated landmark positions from an initial face shape. Despite the success of the methods for near-frontal face alignment, their performance was dramatically degraded on challenging benchmarks. The main reason is that the methods use handcrafted features and simply learned regression methods, which are weak to take full advantage of data for the accurate shape mapping on unconstrained faces.

With the development of deep learning on computer vision, convolutional neural network (CNN)-based methods have achieved impressive performance for unconstrained face alignment. Most existing works focus on improving the accuracy of the landmark localization by utilizing large backbone networks (e.g., VGG-16 [23], ResNet-50/152 [24], and Hourglass [25]). Although the networks have powerful feature extraction ability, they involve many parameters and a high computational cost and are difficult to apply in resource-limited environments.

Recently, some researchers have tended to balance the accuracy and efficiency of a facial landmark detector. They either train a small model from scratch [26,27] or use knowledge distillation (KD) for model compression [28,29,30,31]. The former aims to design a lightweight network combined with an effective learning strategy, while the latter considers how to apply the KD technique to transfer the dark knowledge from a large network to a small one. However, the methods are not flexible enough to adapt to different computing resources as they usually rely on a fixed and carefully designed network structure.

Inspired by the works [32,33] of neural architecture search and neural network pruning for image classification, in which a compact target network was derived from a large supernetwork, we attempted to search for a lightweight face alignment network from a dynamically learned neural architecture. Concretely, we first trained a high-resolution supernetwork based on the structure of HRNet [34]. In this network, a lightweight selective feature fusion (LSFF) block was designed to quantify the importance of the built-in transformation and aggregation operations. Then, we optionally pruned the redundant operations or even the entire blocks to obtain a slimmed network. To reduce the performance gap between the slimmed network and the supernetwork, we developed a triple knowledge distillation scheme, where two peer student networks with masked inputs could learn the ensemble of landmark distributions while receiving the knowledge from a frozen teacher network. In this paper, our main contributions are summarized as follows:We propose a flexible network-level architecture slimming method that can quantify and reduce the redundancy of the network structure to obtain a lightweight facial landmark detector adapted to different computing resources.We design a triple knowledge distillation scheme, in which a slimmed network could be improved without additional complexity by jointly learning the implicit landmark distribution from a teacher network and two peer student networks.Extensive experimental results on challenging benchmarks demonstrate that our approach achieves a better trade-off between accuracy and efficiency than recent state-of-the-art methods (see Figure 1).

The rest of this paper is organized as follows: Section 2 provides a review of related works about existing face alignment methods. In Section 3, we describe the detail of our proposed slimming and distillation methods. Section 4 shows the experimental results and analysis on common datasets. Finally, we give a brief conclusion in Section 5.

## 2. Related Work

In this section, we provide a detailed review of the related methods on face alignment.

### 2.1. Conventional Face Alignment

In the early literature [6,7,8,9], the cascaded regression method was popular and widely used to predict facial landmark positions by resolving a regression problem. The representative methods included SDM [6], ESR [7], LBF [8], and CFSS [9]. The main differences among the methods were the choices of extracted features and the landmark regression methods. SDM used the scale-invariant feature transform (SIFT) as a feature descriptor applied to a cascaded linear regression model. ESR was a two-stage boosted regression method to predict the landmark coordinates by using the shape-indexed features. LBF combined the random forest algorithm with local binary features to accelerate the landmark localization process. To avoid a local optimum due to poor initialization, CFSS exploited hybrid image features to estimate the landmark positions in a coarse-to-fine manner. These methods were weak to detect landmarks on unconstrained face images due to the use of handcrafted features and simply learned regression methods. In our work, we build a CNN model to jointly learn the deep feature extraction and facial landmark heatmap regression.

### 2.2. Large CNN-Based Face Alignment

In recent years, there have been some advanced approaches reported in the literature [10,11,12,13,14,15,16,17,18,19], which have exploited large CNN models to drastically improve the landmark localization accuracy. Wu and Yang [10] proposed a deep variation leveraging network (DVLN), which contained two strongly coupled VGG-16 networks for landmark prediction and candidate decision. Lin et al. [11,12] adopted a classic two-stage detection architecture [35] based on the VGG-16 backbone for joint face detection and alignment. Feng et al. [13] and Dong et al. [14] applied the ResNet-50 [24] and ResNet-152 [24] networks, respectively, as the feature extraction module in the landmark detection process. The stacked hourglass network [25] is a popular CNN backbone used in recent state-of-the-art works [15,16,18] to generate features with multiscale information. Xia et al. [19] combined the HRNet backbone with a transformer structure to achieve a coarse-to-fine face alignment framework. The methods had high accuracy on challenging benchmarks, but inevitably required a large number of parameters and a high computational cost. Our approach only utilizes the large CNN model (HRNet) as a teacher network and adopts a lightweight model for face alignment.

### 2.3. Lightweight CNN-Based Face Alignment

Due to the limited application of large CNN models, some researchers have begun to study the lightweight network design for face alignment. Bulat et al. [26] applied the network quantization technique to construct a binary hourglass network. Guo et al. [27] trained a lightweight network consisting of the MobileNetV2 [36] blocks by using an auxiliary 3D pose estimator. To utilize the learning ability of large models, some recent works [28,29,30,31] used the teacher-guided KD technique to make a small student network learn the dark knowledge from a large teacher network. The student networks were usually based on the existing lightweight networks (e.g., MobileNetV2, EfficientNet-B0 [37], and HRNetV2-W9 [34]), while the teacher networks use the large CNN models (e.g., ResNet-50, EfficientNet-B7 [37], and HRNetV2-W18 [34]) as the network backbone. It is worth mentioning that the KD technique was also applied to improve a facial landmark detector [38,39,40] by mining the spatial–temporal relation from unlabeled video data. Inspired by the student-guided KD [41] that made student networks learn from each other without a teacher network, we introduce a student-guided learning strategy into the original KD framework, which can generate more robust supervision knowledge for learning landmark distribution. Moreover, our student network is derived from a supernetwork and thus has a more flexible structure than other handcrafted models.

## 3. Methods

As illustrated in Figure 2, our approach is a two-stage process consisting of a network-level architecture slimming and triple knowledge distillation, which results in a lightweight facial landmark detector.

### 3.1. Network-Level Architecture Slimming

Our high-resolution supernetwork (HRSuperNet) follows a similar structure to HRNet in Figure 3 and begins from a stem that is composed of two 3×3 convolutions with a stride of 2. The spatial resolution is downsampled to H/4×W/4, where *H* and *W* denote the height and width of an input image $I\in R^{3\times H\times W}$. The main body consists of ten stages maintaining the high-resolution representations throughout the network. Different from HRNet, the supernetwork contains a single-resolution LSFF block with a downsampling ratio of 1 in the first stage and repeats four-resolution blocks with downsampling ratios of {1, 1/2, 1/4, 1/8} from the beginning of the second stage. Each block has four stacked mobile inverted bottleneck convolutions (MBConvs [36]) with a 3×3 kernel size and an expansion ratio of 1. The design could make the supernetwork keep a larger architecture space but fewer parameters and lower computational cost than HRNet. Except for the first stage, the LSFF block is designed to transform and aggregate features from the previous stage and generate new features as inputs to the next stage. The process is formulated as follows:(1)$$\begin{array}{ccc} \; & Y_{i>1,k}=E\left(\sum_{j=1}^{J_{i-1}}\alpha_{i,j}^{k}T\left(X_{i-1,j}^{k}\right)\right), & \end{array}$$
where $Y_{i,k}$ is the output of the *k*th block in the *i*th stage and $X_{i-1,j}^{k}$ denotes the *k*th output from the *j*th block in the (i−1)th stage. $J_{i-1}$ is the number of blocks in the (i−1)th stage. *T* represents a transformation operation that is either a 1×1 convolution with a stride of 1 and a bilinear interpolation for upsampling, a sequence of 3×3 convolutions with a stride of 2 for downsampling, or an identity shortcut connection. *E* denotes a feature encoding operation implemented by the stacked MBconvs. The factor α is used as the weight of each transformation operation to participate in the follow-up aggregation process. The head in the supernetwork consists of two 1×1 convolutions with a stride of 1 and generates the landmark heatmaps $P\in R^{N\times M\times H/4\times W/4}$ when receiving *N* samples with *M* facial landmark points.

During the training, we make the supernetwork learn the landmark heatmap regression along with the subnetwork architecture search by imposing an L1 regularization on α to enforce the sparsity of the operations with few contributions to the network. Formally, the overall training loss is:(2)$$L=\sum_{n=1}^{N}\sum_{m=1}^{M}\frac{MSE(P_{n,m},G_{n,m})}{N\times M}+\lambda \sum_{i=2}^{I}\sum_{j=1}^{J_{i-1}}\sum_{k=1}^{J_{i}}\left|\alpha_{i,j}^{k}\right|,$$
where $MSE(P_{n,m},G_{n,m})$ denotes the standard mean square error between the predicted heatmap $P_{n,m}$ and the ground-truth heatmap $G_{n,m}$ of the *m*th landmark in the *n*th sample. The ground-truth heatmap is generated by applying a 2D Gaussian centered on the ground-truth location of each landmark. λ is the weight to balance the MSE and the L1 penalty term. *I* and $J_{i}$ denote the number of stages and the number of blocks in the *i*th stage, respectively. We first train the supernetwork by alternately optimizing the importance factors and the network weights until they converge. Then, we prune the redundant transformation and aggregation operations in the LSFF blocks, where the corresponding factors are smaller than a given pruning threshold. Note that the entire block is discarded if all the associated operations are pruned.

### 3.2. Triple Knowledge Distillation

In our distillation scheme, we adopt the slimmed network as the peer student networks $S_{1}$ and $S_{2}$ and use the pretrained HRNet as the teacher network *T*. To increase the model diversity, we use the occluded images with a random-sized mask as the inputs of the student networks.

Specially, we define a KD loss for a network to learn the landmark distribution from another network as follows:(3)$$L_{KD}(P^{2}\left|\right|P^{1})=\sum_{n=1}^{N}\sum_{m=1}^{M}\frac{D_{KL}(S\left(P_{n,m}^{2}\right)||S\left(P_{n,m}^{1}\right))}{N\times M},$$
where $D_{KL}$ is the Kullback–Leibler (KL) divergence to measure the distance of the landmark distributions from $S(P^{1})$ to $S(P^{2})$, and *S* is the softmax function working on the predicted landmark heatmaps $P^{1}$ and $P^{2}$.

During the training, we use MSE and $L_{KD}$ as the main criterion to make the student networks learn the explicit landmark distribution from the ground-truth heatmap, while allowing them to learn the implicit landmark distribution from their ensemble predictions and the output of the teacher network. The overall training loss of a student network $S_{i}$ is formulated as:(4)$$\begin{array}{cc} P^{E} & =(P^{S_{1}}+P^{S_{2}})/2,\\ L_{S_{i}} & =\sum_{n=1}^{N}\sum_{m=1}^{M}\frac{MSE(P_{n,m}^{S_{i}},G_{n,m})}{N\times M}\\ & +\lambda_{1}L_{KD}(P^{E}\left|\right|P^{S_{i}})+\lambda_{2}L_{KD}(P^{T}\left|\right|P^{S_{i}}), \end{array}$$
where $P^{S_{1}}$, $P^{S_{2}}$, and $P^{T}$ denote the predicted landmark heatmaps of $S_{1}$, $S_{2}$, and *T*, respectively. The weights $\lambda_{1}$ and $\lambda_{2}$ are used to balance MSE and the KD losses.

## 4. Experiments

### 4.1. Datasets

We conducted experiments on three challenging datasets including 300W [20], COFW [21], and WFLW [15].

300W: It consists of the HELEN [42], LFPW [43], AFW [44], XM2VTS [45], and IBUG [20] datasets, where each face has 68 landmarks. The training set contains 3148 images and the test set has 689 images divided into the challenge subset (135 images) and the common subset (554 images). Masked 300W [46] is a supplement to the 300W dataset for testing. This dataset mainly includes masked faces with over 50% of occlusion.

COFW: It contains 1852 face images with different degrees of occlusion including 1345 training images and 507 test images. Each face image has 29 annotated landmarks.

WFLW: There are 7500 images for training and 2500 images for testing where the test set includes six subsets: large pose (326 images), illumination (698 images), occlusion (736 images), blur (773 images), make-up (206 images), and expression (314 images).

### 4.2. Evaluation Metrics

We followed previous works and used the normalized mean error (NME) to evaluate the performance of the facial landmark detection:(5)$$NME=\sum_{n=1}^{N}\sum_{m=1}^{M}\frac{{∥p_{n,m}-g_{n,m}∥}_{2}}{N\times M\times d}$$
where $p_{n,m}$ and $g_{n,m}$ denote the coordinate vectors of the predicted landmark and the ground-truth landmark, respectively. *d* is the interocular distance. We also report the failure rate by setting a maximum NME of 10%. The number of parameters (#Params) and the number of floating-point operations (FLOPs) were used to measure model size and computational cost, respectively.

### 4.3. Implementation Detail

Following the work [15], all the faces were cropped based on the provided bounding boxes and resized to 256×256. We augmented the data by a 1.0±0.25 scaling, ±30-degree rotation, and random flipping with a probability of 50%. The pseudocode in Algorithm 1 shows the training pipeline of our approach in the slimming and distilling stages.
**Algorithm 1: **SD-HRNet Algorithm
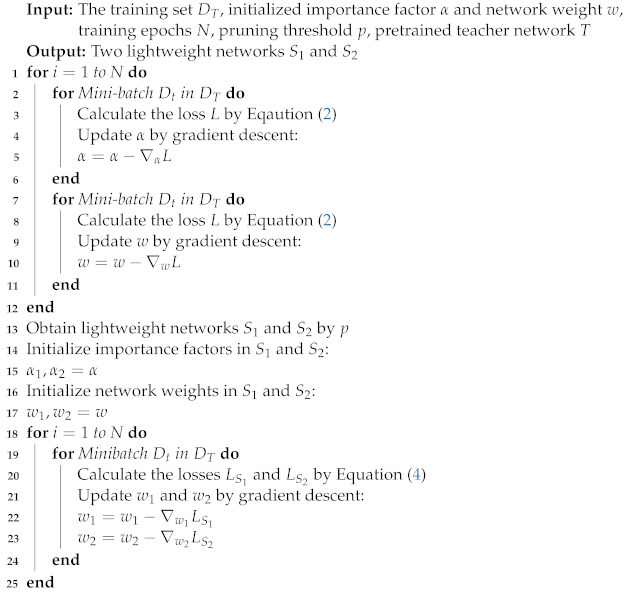


Slimming stage: We alternatively optimized the importance factors and network weights for 60 epochs. To optimize the importance factors, we used the Adam optimizer with the learning rates of $1.8\times 10^{-4}$ on 300W and WFLW, and $3.5\times 10^{-4}$ on COFW. The weight λ was set to $5\times 10^{-5}$. To update the network weights, we used the Adam optimizer with a momentum of 0.9 and weight decay of $4\times 10^{-5}$. The initial learning rate was $1\times 10^{-4}$ on 300W and COFW, and $2\times 10^{-4}$ on WFLW, which was dropped by a factor of 0.1 in 40 and 50 epochs. The pruning threshold was set to 0.0017 on 300W, 0.0042 on COFW, and 0.002 on WFLW. The batch size was set to 16 on 300W and COFW, and 32 on WFLW.

Distilling stage: We jointly fine-tuned two slimmed networks for 60 epochs. The settings of the optimizer, learning rate and batch size were the same as those in the slimming stage. The weights $\lambda_{1}$ and $\lambda_{2}$ were set to 4 and 1 on 300W, 3 and 0.1 on COFW, and 0.3 and 0.1 on WFLW.

### 4.4. Comparison with State-of-the-Art Methods

In this section, we compare our approach with the recent state-of-the-art methods on 300W, COFW, and WFLW. S-HRNet is the slimmed network from HRSuperNet. SD-HRNet${}_{1}$ and SD-HRNet${}_{2}$ denote two refined S-HRNets through the proposed distillation scheme. We report the average results with the standard deviation of SD-HRNet${}_{1}$ and SD-HRNet${}_{2}$ from training them five times over different seeds.

#### 4.4.1. Results on 300W

We report the #Params and FLOPs in Table 1 as well as the NME on the 300W subsets in Table 2. Compared to the advanced models (e.g., LAB and SLPT) with large backbones, our method (SD-HRNet) had far fewer parameters and FLOPs while achieving competitive or even better performance. Compared to HRNet, SD-HRNet only increased the NME by about 2% but reduced the #Params by 89.5% and the FLOPs by 86.3%. We showed that SD-HRNet achieved fewer parameters (0.98 M parameters) and a lower computational cost (0.59 G FLOPs) than existing lightweight models. Moreover, we proved the effectiveness of the proposed slimming and distillation approaches as the #Params and FLOPs of HRSuperNet were reduced by 70.0% and 52.8%, respectively, and the NME of S-HRNet was reduced by about 3%. Table 3 shows the performance of the methods on Masked 300W. We found that SD-HRNet had an obvious improvement for occluded faces due to the introduction of masked inputs. Although our method obtained a competitive NME compared to most previous methods, it underperformed the recent state-of-the-art methods [47,48] focusing on the occlusion problem.

#### 4.4.2. Results on COFW

Table 4 shows the NME and the failure rate for a maximum NME of 10% on the COFW test set. Our method performed better than some classic works (e.g., RAR and DAC-CSR) for partially occluded face alignment and achieved a competitive accuracy against recent state-of-the-art methods (e.g., LGSA and SLPT). Compared to HRNet, the NME of SD-HRNet was slightly increased by about 5% while the #Params and FLOPs were reduced by 88.7% and 86.0%, respectively. Moreover, SD-HRNet was still more lightweight than recent small models and had similar landmark detection performance.

#### 4.4.3. Results on WFLW

In Table 5, we report the NME on the WFLW test set and six subsets. Our method significantly outperformed conventional cascaded regression methods (e.g., SDM and CFSS) and some classic large models (e.g., LAB and Wing+PDB). However, we found that there was a slightly bigger performance gap between SD-HRNet and recent large models than the results on 300W and COFW. The reason might be that learning the dense landmark regression relies on a large network capacity. Compared to the advanced lightweight models, SD-HRNet achieved the third-best NME in most cases with a better trade-off of model size (1.32 M parameters) and computational cost (0.6 G FLOPs).

In Figure 4, we show the number of frames per second (FPS) of our method using different batch sizes on 300W. Due to the very small resource consumption, SD-HRNet could process more than 500 samples per second. In Figure 5, we give some example results on the common datasets and show the accurate landmark localization of our method on various unconstrained faces. All the experiments were implemented with PyTorch on a single TITAN Xp GPU.

### 4.5. Ablation Study

In this section, we conduct an ablation study on 300W and analyze the effect of the proposed components.

#### 4.5.1. HRNet vs. HRSuperNet

To verify the rationality of our supernetwork, we trained HRNet and HRSuperNet on 300W without pretraining and used them as the supernetwork to generate a series of slimmed networks. The original residual units [34] or stacked MBConvs were used as the feature encoding operation in the proposed LSFF block. As seen from Figure 6, most networks derived from HRSuperNet had a lower NME than HRNet when their FLOPs were similar, which suggested that a larger architecture space was more likely to generate better subnetworks.

#### 4.5.2. KD Components

In Table 6, we show the effect of different KD components in our distillation scheme for the performance on 300W. We observed that each component incrementally led to the improvement of the slimmed network. It suggested that the combination of the teacher-guided KD and the student-guided KD was an effective way for the implicit knowledge transfer. In addition, the introduction of masked inputs could increase the diversity of student networks and make them learn robust landmark distribution from each other.

### 4.6. Visualization of the Architectures

We visualize the slimmed architecture trained on 300W in Figure 2 and the other two architectures on COFW and WFLW in Figure 7. The proposed selective feature fusion mechanism could result in different network structures from a unified architecture space, which were adapted to different datasets and landmark detection tasks. For example, the architectures from 300W and COFW tended to preserve more high-resolution blocks from the first and second branches than the architecture from WFLW. In addition, we found that more than 94% of the blocks in HRSuperNet were utilized by the slimmed architectures. It suggested that the designed architecture space was reasonable to cover most cases for generating an efficient face alignment network.

## 5. Conclusions

In this paper, we proposed a network-level slimming method and a hybrid knowledge distillation scheme, which could work together to generate an efficient and accurate facial landmark detector. Compared to existing handcrafted models, our model achieved competitive performance with a better trade-off between model size (0.98 M–1.32 M parameters) and computational cost (0.59 G–0.6 G FLOPs). In addition, our method was more flexible in practical application through an adaptive architecture search technique, which could be applied to real-time human–computer interaction systems under different resource-limited environments. Nevertheless, there was still a performance gap between our method and recent state-of-the-art large models, especially for the dense or strongly occluded landmark detection task. In future work, we will explore how to design a more reasonable architecture search space to improve the upper bound of performance and extend our method to other computer vision tasks such as human pose estimation and semantic segmentation.

## Figures and Tables

**Figure 1 sensors-23-01532-f001:**
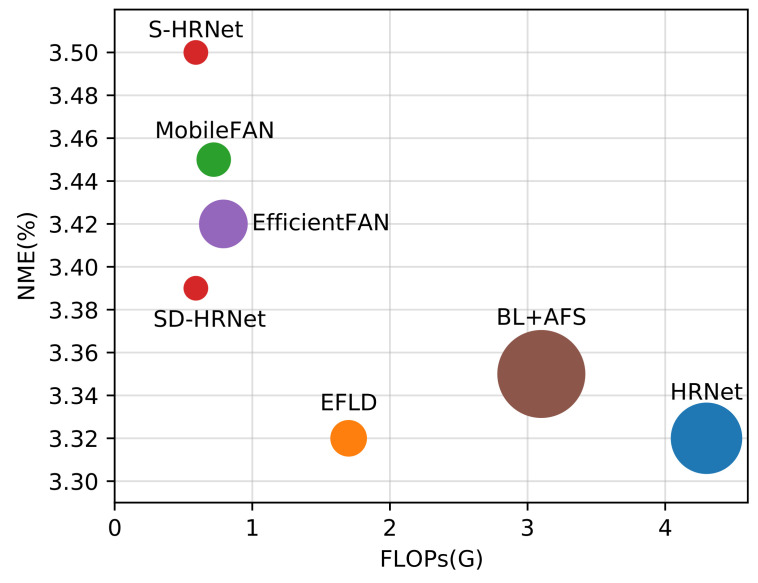
Comparison of the computational cost (i.e., FLOPs) and the performance (i.e., NME) on 300W between the proposed approach and existing state-of-the-art methods. The size of a circle represents the number of parameters. Our approach (SD-HRNet) achieves a better trade-off between accuracy and efficiency than its counterparts.

**Figure 2 sensors-23-01532-f002:**
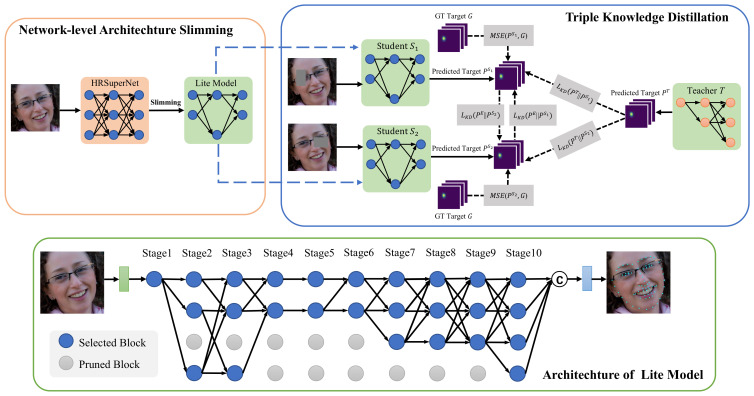
Illustration of the proposed slimming and distillation procedures for face alignment. A lightweight model is first obtained by slimming the HRSuperNet. Then, the lightweight model is refined in a triple knowledge distillation scheme consisting of two peer student networks and a teacher network. We visualize the architecture of the lightweight model trained on the 300W dataset, where the redundant TA operations and LSFF blocks are pruned.

**Figure 3 sensors-23-01532-f003:**
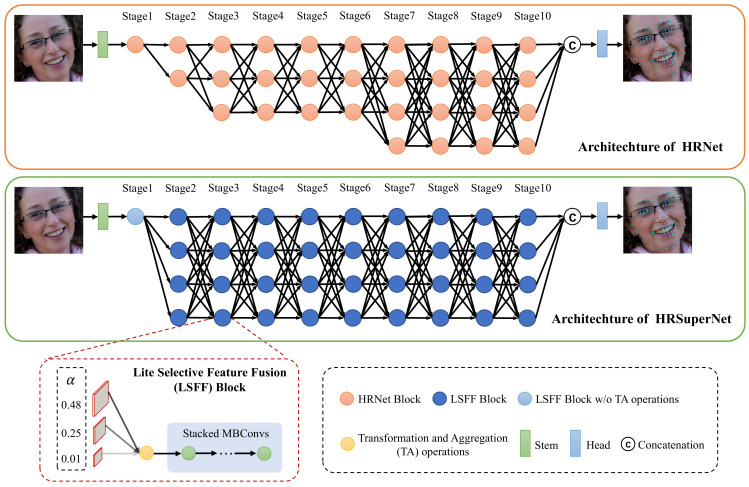
Detailed structures of HRNet and HRSuperNet. The proposed lightweight selective feature fusion (LSFF) block is composed of the transformation and aggregation (TA) operations with different importance factors α and stacked mobile inverted bottleneck convolutions (MBConvs).

**Figure 4 sensors-23-01532-f004:**
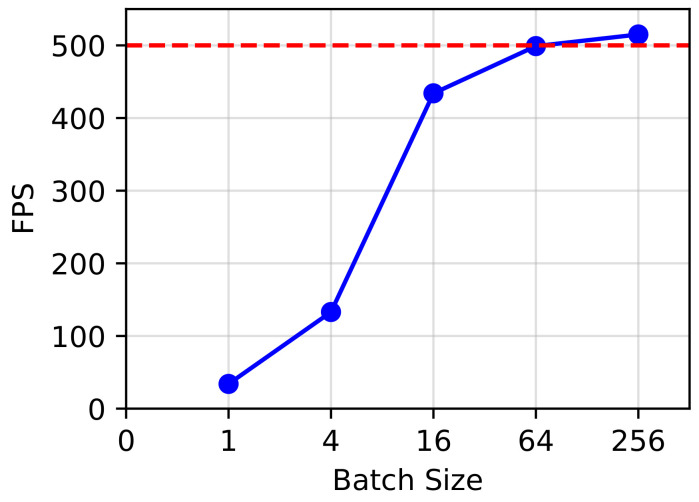
FPS of our method using different batch sizes on 300W.

**Figure 5 sensors-23-01532-f005:**
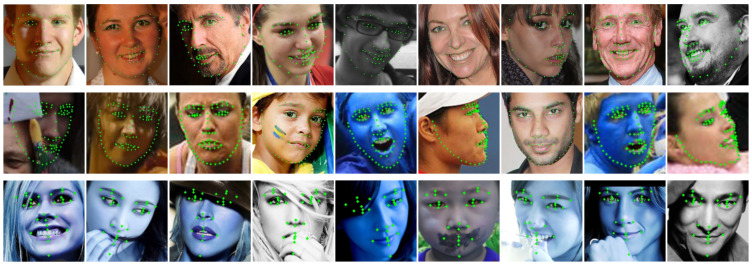
Example results of our method for face alignment. Top row: results on 300W (68 points). Second row: results on WFLW (98 points). Bottom row: results on COFW (29 points).

**Figure 6 sensors-23-01532-f006:**
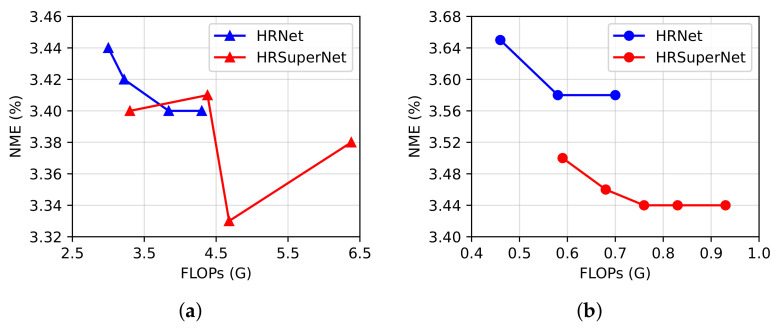
Comparison of HRNet and HRSuperNet based on original residual units (**a**) or stacked MBConvs (**b**), which were used as the supernetwork in the proposed slimming method. We obtained a series of slimmed networks with different NME and FLOPs on the 300W full set by using different pruning thresholds.

**Figure 7 sensors-23-01532-f007:**
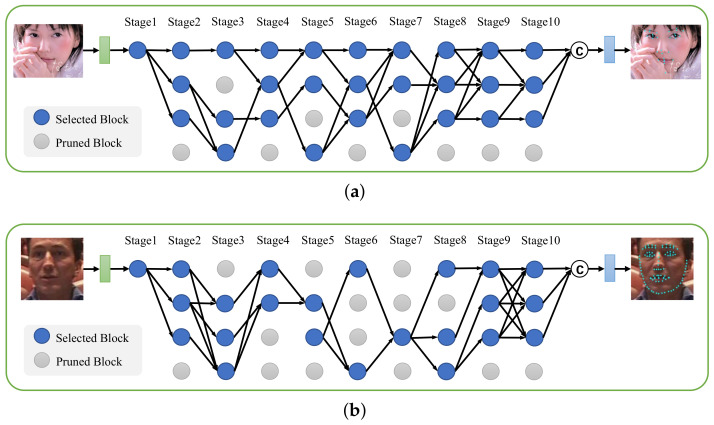
Visualization of the slimmed architectures trained on the COFW (**a**) and WFLW (**b**) datasets.

**Table 1 sensors-23-01532-t001:** Comparison of different methods in backbone, #Params, and FLOPs.

Method	Backbone	#Params (M)	FLOPs (G)
DVLN [10]	VGG-16 [23]	132.0	14.4
Wing+PDB [13]	ResNet-50 [24]	25	3.8
SAN [14]	ResNet-152 [24]	57.4	10.7
LAB [15]	Hourglass [25]	25.1	19.1
HRNet [34]	HRNetV2-W18 [34]	9.3	4.3
AWing [16]	Hourglass [25]	24.15	26.79
BL+AFS [17]	-	14.29	3.10
LGSA [18]	Hourglass [25]	18.64	15.69
SLPT [19]	HRNetV2-W18 [34]	13.18	5.17
SRN [47]	Hourglass [25]	19.89	-
GlomFace [48]	-	-	13.48
MobileFAN [28]	MobileNetV2 [36]	2.02	0.72
EfficientFAN [29]	EfficientNet-B0 [37]	4.19	0.79
EFLD [30]	HRNetV2-W9 [34]	2.3	1.7
mnv2${}_{\mathrm{KD}}$ [31]	MobileNetV2 [36]	2.4	0.6
HRSuperNet	HRSuperNet	3.27	1.25
SD-HRNet (300W)	S-HRNet	0.98	0.59
SD-HRNet (COFW)	S-HRNet	1.05	0.60
SD-HRNet (WFLW)	S-HRNet	1.32	0.60

**Table 2 sensors-23-01532-t002:** Comparison of NME (%) on 300W: common subset, challenge subset, and full set.

Method	Common	Challenge	Full
ODN [49]	3.56	6.67	4.17
SAN [14]	3.34	6.60	3.98
LAB [15]	2.98	5.19	3.49
HRNet [34]	2.87	5.15	3.32
AWing [16]	2.72	4.52	3.07
BL+AFS [17]	2.89	5.23	3.35
LGSA [18]	2.92	5.16	3.36
SAAT [46]	2.87	5.03	3.29
SRN [47]	3.08	5.86	3.62
GlomFace [48]	2.79	4.87	3.20
SLPT [19]	2.75	4.90	3.17
MobileFAN [28]	2.98	5.34	3.45
EfficientFAN [29]	2.98	5.21	3.42
EFLD [30]	2.88	5.03	3.32
mnv2${}_{\mathrm{KD}}$ [31]	3.56	6.13	4.06
HRSuperNet	3.00	5.28	3.45
S-HRNet	3.02	5.44	3.50
SD-HRNet${}_{1}$	2.93 ± 0.01	5.32 ± 0.05	3.40 ± 0.01
SD-HRNet${}_{2}$	2.94 ± 0.01	5.33 ± 0.02	3.41 ± 0.01

**Table 3 sensors-23-01532-t003:** Comparison of NME (%) on Masked 300W: common subset, challenge subset, and full set.

Method	Common	Challenge	Full
CFSS [9]	11.73	19.98	13.35
DHGN [50]	8.98	12.19	9.61
SBR [38]	8.72	13.28	9.6
SHG [25]	8.17	13.52	9.22
MDM [51]	7.66	11.67	8.44
FHR [52]	7.02	11.28	7.85
LAB [15]	6.07	9.59	6.76
SAAT [46]	5.42	11.36	6.58
SRN [47]	5.78	9.28	6.46
GlomFace [48]	5.29	8.81	5.98
HRSuperNet	14.03	20.52	15.30
S-HRNet	19.18	29.37	21.17
SD-HRNet${}_{1}$	6.43 ± 0.25	11.05 ± 0.51	7.34 ± 0.28
SD-HRNet${}_{2}$	6.23 ± 0.23	10.72 ± 0.25	7.11 ± 0.20

**Table 4 sensors-23-01532-t004:** Comparison of NME (%) and failure rate (%) for a maximum NME of 10% on the COFW test set.

Method	NME (%)	Failure Rate (%)
HPM [53]	7.50	13.00
CCR [54]	7.03	10.9
DRDA [55]	6.46	6.00
RAR [56]	6.03	4.14
DAC-CSR [57]	6.03	4.73
Wing+PDB [13]	5.07	3.16
LAB [15]	3.92	0.39
HRNet [34]	3.45	0.19
LGSA [18]	3.13	0.002
SLPT [19]	3.32	0.00
MobileFAN [28]	3.66	0.59
EfficientFAN [29]	3.40	0.00
EFLD [30]	3.50	0.00
mnv2${}_{\mathrm{KD}}$ [31]	4.11	2.36
HRSuperNet	3.74	0.59
S-HRNet	3.69	0.20
SD-HRNet${}_{1}$	3.61 ± 0.02	0.12 ± 0.16
SD-HRNet${}_{2}$	3.63 ± 0.03	0.20 ± 0.17

**Table 5 sensors-23-01532-t005:** Comparison of NME (%) on the WFLW test set and 6 subsets: pose, expression, illumination, make-up, occlusion, and blur.

Method	Test	Pose	Expression	Illumination	Make-Up	Occlusion	Blur
ESR [58]	11.13	25.88	11.47	10.49	11.05	13.75	12.20
SDM [6]	10.29	24.10	11.45	9.32	9.38	13.03	11.28
CFSS [9]	9.07	21.36	10.09	8.30	8.74	11.76	9.96
DVLN [10]	6.08	11.54	6.78	5.73	5.98	7.33	6.88
LAB [15]	5.27	10.24	5.51	5.23	5.15	6.79	6.32
Wing+PDB [13]	5.11	8.75	5.36	4.93	5.41	6.37	5.81
HRNet [34]	4.60	7.94	4.85	4.55	4.29	5.44	5.42
AWing [16]	4.36	7.38	4.58	4.32	4.27	5.19	4.96
LUVLi [59]	4.37	7.56	4.77	4.30	4.33	5.29	4.94
LGSA [18]	4.28	7.63	4.33	4.16	4.27	5.33	4.95
mnv2${}_{\mathrm{KD}}$ [31]	8.57	15.06	8.81	8.15	8.75	9.92	9.40
MobileFAN [28]	4.93	8.72	5.27	4.93	4.70	5.94	5.73
EFLD [30]	4.74	8.41	5.01	4.71	4.57	5.70	5.45
EfficientFAN [29]	4.54	8.20	4.87	4.39	4.54	5.42	5.04
HRSuperNet	4.83	8.45	5.10	4.80	4.85	5.79	5.53
S-HRNet	4.98	8.68	5.33	4.86	4.88	5.87	5.70
SD-HRNet${}_{1}$	4.93 ± 0.01	8.63 ± 0.03	5.31 ± 0.05	4.81 ± 0.03	4.76 ± 0.02	5.73 ± 0.02	5.56 ± 0.03
SD-HRNet${}_{2}$	4.96 ± 0.03	8.66 ± 0.10	5.35 ± 0.04	4.82 ± 0.05	4.81 ± 0.04	5.76 ± 0.04	5.61 ± 0.06

**Table 6 sensors-23-01532-t006:** NME (%) of our method using different KD components on the 300W full set.

Teacher	Peer Student	Masked Inputs	NME (%)
✖	✖	✖	3.50
✓	✖	✖	3.46
✓	✓	✖	3.44
✓	✓	✓	3.39

## Data Availability

The code is available at https://github.com/MUST-AI-Lab/SD-HRNet (accessed on 27 December 2022).

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
