# Peer review of "SD-HRNet: Slimming and Distilling High-Resolution Network for Efficient Face Alignment"

_sensors, 2023, doi:10.3390/s23031532_

Round 1

Reviewer 1 Report

The authors presented in this paper a network-level slimming method and a hybrid knowledge distillation scheme, which could work together to generate an efficient and accurate facial landmark detector.

-          The paper is well-written, and the study is well-planned.

-          Background information is thoroughly presented, and the paper's contribution is interesting.

-          Overall, this is a good paper and contributes to the science as it proposes an interesting approach to resolving a challenging problem.

However:

-          The paper’s organization should be added in the last part of the introduction.

-          What are the practical implications of your research?

-          What are the limitations of the present work?

-          Uncertainties of the model should be reported.

-          Add a list of abbreviations used in the manuscript.

-          Authors should quantify (i.e., highlight the main results) in the abstract and conclusion sections.

-          The below papers has some interesting implications that you could discuss in your Introduction and how it relates to your work: https://doi.org/10.3390/electronics9081188 and https://doi.org/10.1016/j.dsp.2020.102809

Reviewer 2 Report

The authors proposed a network-level architecture slimming method called SD-HRNet. Extensive experiments demonstrated that the proposed method achieved competitive performance. However, there are still some concerns that need to be addressed before publication.

1.The writing of the manuscript needs to be further polished. The authors should carefully check the manuscript for writing and conceptual description.

2.Triple Knowledge Distillation uses masking blocks to augment the data. Therefore, the authors should test their model on the masked 300W[1].

3. The authors should analyse whether the high-resolution hyper-segmentation network is robust to obscured data. Please cite and compare recent works [2,3,4,5] focusing on the occlusion problem.

4. One of the core contributions of this work is a new knowledge distillation module. Therefore, in Related work, the authors should introduce some recent works on knowledge distillation and their application in face alignment tasks [6] [7] [8].

[1] Improving robustness of facial landmark detection by defending against adversarial attacks

[2] Occlusion-Robust Face Alignment Using a Viewpoint-Invariant Hierarchical Network Architecture

[3] Robust facial landmark detection via occlusion adaptive deep networks

[4] Reasoning structural relation for occlusion-robust facial landmark localization

[5] Occlusion coherence: Detecting and localizing occluded faces.

[6] Supervision-by-registration: An unsupervised approach to improve the precision of facial landmark detectors

[7] Towards Omni-Supervised Face Alignment for Large Scale Unlabeled Videos

[8] Multi-sourced Knowledge Integration for Robust Self-Supervised Facial Landmark Tracking

Round 2

Reviewer 1 Report

The authors have adressed all my previous comments. The manuscript can be accepted in the current form.